# An Improved Reference Gene for Detection of “*Candidatus* Liberibacter asiaticus” Associated with Citrus Huanglongbing by qPCR and Digital Droplet PCR Assays

**DOI:** 10.3390/plants10102111

**Published:** 2021-10-05

**Authors:** Manjunath L. Keremane, Thomas G. McCollum, Mikeal L. Roose, Richard F. Lee, Chandrika Ramadugu

**Affiliations:** 1USDA ARS National Clonal Germplasm Repository for Citrus and Dates, Riverside, CA 92507, USA; manjunath.keremane@usda.gov (M.L.K.); rfleevirus@comcast.net (R.F.L.); 2USDA ARS Horticultural Research Laboratory, Fort Pierce, FL 34945, USA; greg.mccollum@usda.gov; 3Department of Botany and Plant Sciences, University of California Riverside, Riverside, CA 92521, USA; Mikeal.roose@ucr.edu

**Keywords:** huanglongbing, citrus, reference gene, malate dehydrogenase, qPCR, digital PCR

## Abstract

Citrus huanglongbing (HLB) disease associated with the ‘*Candidatus* Liberibacter asiaticus’ (CLas) bacterium has caused significant financial damage to many citrus industries. Large-scale pathogen surveys are routinely conducted in California to detect CLas early in the disease cycle by lab-based qPCR assays. We have developed an improved reference gene for the sensitive detection of CLas from plants in diagnostic duplex qPCR and analytical digital droplet PCR (ddPCR) assays. The mitochondrial cytochrome oxidase gene (COX), widely used as a reference, is not ideal because its high copy number can inhibit amplification of small quantities of target genes. In ddPCRs, oversaturation of droplets complicates data normalization and quantification. The variable copy numbers of COX gene in metabolically active young tissue, greenhouse plants, and citrus relatives suggest the need for a non-variable, nuclear, low copy, universal reference gene for analysis of HLB hosts. The single-copy nuclear gene, malate dehydrogenase (MDH), developed here as a reference gene, is amenable to data normalization, suitable for duplex qPCR and ddPCR assays. The sequence of MDH fragment selected is conserved in most HLB hosts in the taxonomic group Aurantioideae. This study emphasizes the need to develop standard guidelines for reference genes in DNA-based PCR assays.

## 1. Introduction

Citrus huanglongbing (HLB or citrus greening disease) is associated with an alphaproteobacterium, “*Candidatus* Liberibacter asiaticus” (CLas), and vectored by *Diaphorina citri* Kuwayama (Asian citrus psyllid; ACP) [1,2,3]. HLB has caused significant financial damage to citrus industries worldwide. In California (CA), HLB was first reported in 2012 in a suburban area of Los Angeles [4]. Since then, intensive efforts have been in place to detect, eradicate, and contain the spread of HLB. As of August 2021, over 2400 HLB positive trees have been found, mostly in non-commercial settings in Southern California, posing a serious threat to the $2.4-billion citrus industry.

In-depth genetic analyses of CLas isolates characterized from many locations in CA show differences in sequence characteristics and suggest multiple introductions of the pathogen [5]. HLB finds in CA are monitored by intensive follow-up surveys, aggressive vector control, eradicating infected trees, and other quarantine measures. While the disease has spread widely in certain locations, in other areas in CA, the HLB-positive trees and CLas-positive ACP were restricted to a very small number of plants and/or insects with no apparent secondary spread from the foci of infection. Early detection of the disease is essential for timely management strategies essential for the survival of the CA citrus industry. In CA, HLB surveys are conducted by testing ACP, since they indicate disease incidence early in an area [6]. Subsequent intensive testing of surrounding trees is conducted using mature leaves [7] and young flush where ACP feeds, acquires, oviposits, and transmits the HLB pathogen [8]. As a result of the long pathogen incubation period in citrus trees infected with CLas [9], identification of CLas-infected but asymptomatic trees with low levels of the HLB-associated pathogen is essential for the prevention of the further spread of the pathogen.

Standard testing methods for CLas from citrus involve duplex real-time PCR assays (qPCR) for the simultaneous detection of a mitochondrial cytochrome oxidase gene (COX) and the 16S rDNA of CLas [7]. Under certain conditions, COX may not be optimal as a reference gene for inclusion in diagnostic assays. Mitochondria are generally present at high copy numbers in host cells. The numbers can vary depending on multiple factors such as host species, tissue type, growing conditions, physiological status, etc. [10,11]. In extractions containing low titers of CLas, a high copy reference gene may interfere with the detection of the target pathogen gene. In gene expression studies by qPCR, selecting a reference gene that is expected to have a cycle threshold value closer to the gene of interest is preferred [12,13]. In HLB diagnostic assays, results from extractions containing low amounts of pathogen or host DNA may also lead to the assumption of false negatives, a situation that is not desirable for pathogen exclusion strategy designed to prevent disease establishment and spread. 

Digital droplet PCR (ddPCR) is an absolute quantification assay that can estimate the number of target molecules in a sample [14]. Detection of citrus pathogens such as *Xanthomonas citri* (associated with citrus canker) and *Spiroplasma citri* (causing citrus stubborn disease) by ddPCR have been described [15,16]. ddPCR assays have been developed for the detection of HLB-associated CLas [17,18]. However, these studies and our earlier ddPCR assays did not include a reference gene essential as an internal control for the normalization of data and correct interpretation of results. The commonly used mitochondrial COX gene is not ideal as a reference gene in ddPCR assays, since the high copy numbers of mitochondria per cell [19,20] compromise the sensitivity of the assay system. In CA, there is a need to modify currently used methods to avoid false negatives that may lead to escape trees which can act as foci for new infections and disease spread. Developing a single/low copy reference gene is also essential for absolute quantification of the target and data normalization in ddPCR [21].

The goals of the current study were: (1) to identify an alternate reference gene that has a low copy number in citrus hosts; (2) to develop assays using a region of the reference gene highly conserved in members of subfamily Aurantioideae (family Rutaceae) so that all known natural plant hosts of HLB can be evaluated in diagnostic assays; (3) comparative analysis of the newly developed reference gene with COX gene in HLB diagnostic assays and; (4) to develop the reference gene suitable for data normalization in qPCR and ddPCR assays, while analyzing plant samples of *Citrus* and other closely related genera. 

## 2. Results

### 2.1. Analysis of a Single Copy Nuclear Gene in Aurantioideae 

In a previous study [22], we analyzed 5632 bp sequences from six nuclear genes from a wide variety of citrus and citrus relatives to identify hybrid origins of the taxa included in the study. The MDH gene fragment included in previous studies is conserved in a wide range of Aurantioid taxa and was evaluated as a possible reference gene. A 900 bp region of the nuclear malate dehydrogenase (MDH) gene was PCR-amplified, cloned, and sequenced from representative citrus cultivars and many related genera as described in materials and methods. A minimum of eight clones per accession were sequenced to document haplotype differences. Sequences from a total of 197 Rutaceous taxa have been deposited in Genbank; accession numbers for 57 representative taxa (a total of 74 haplotypes) are shown in Appendix A. An alignment of the 900 bp region of the amplified region of MDH with 74 sequences representing diverse taxa of family Rutaceae, mostly from the subfamily, Aurantioideae (in addition to two accessions from Rutoideae and one from Flindersioideae) was conducted using ClustalX [23]. A cladogram was constructed using the Maximum Parsimony method using the subtree-pruning-regrafting algorithm (Mega v10.0.5) [24,25]. The different taxonomic sub-groups cluster together, showing sequence similarity in closely related accessions (Figure 1). Many members of Rutaceae can serve as hosts for the HLB pathogen and hence are tested in duplex qPCR assays for the presence of CLas. A gene fragment that is conserved in many citrus accessions and related taxa may be suitable as a reference gene for duplex qPCR assays and normalizing data in digital PCRs. The cladogram shows that the 900 bp region of the single-copy nuclear gene MDH represents the general phylogeny of the group, Aurantioideae.

### 2.2. Primers and Probes for Detecting Internal Control Gene Malate Dehydrogenase

We selected a 92-bp fragment from the 900 bp region of the MDH gene to evaluate as internal control reference gene in qPCR assays. Figure 2 shows the sequence alignment of the 92-bp region of MDH from a small set of individuals representing ten diverse taxa of Aurantioideae and one taxon from Rutoideae subfamilies. The primer sequences and identities are shown. The forward primer spans an intron junction; the 5′ ten bases correspond to an intron, and the rest align to an adjacent exon (Appendix A). The primers were designed to selectively amplify from DNA templates only. A list of all primers and probes used in the study are shown in Table 1.

### 2.3. Comparison of qPCR and ddPCR Assays Using a Plasmid Control for Detection of CLas

A plasmid preparation containing CLas 16S rDNA cloned fragment, estimated to contain 1.2 × 10^9^ copies/µL, was used to prepare a ten-fold serial dilution in 1× TE to determine the linearity and dynamic range of detection in qPCR and digital droplet PCR (ddPCR) assays. In qPCR, the standard dilutions utilized in the experiment were detected over a dynamic range of nine dilutions as expected: 10^−1^ to 10^−9^ (Figure 3A). The cycle threshold values for the amplification plots ranged from 9.1 to 36.2. Aliquots from the same plasmid dilution preparations were used for conducting ddPCR assays and analyzed. In ddPCR, serial dilutions 10^−1^ to 10^−5^ were not informative since most of the droplets were saturated with the target DNA amplicons, and negative droplets (without the target) were either absent or were detected in very low numbers. However, in dilutions 10^−6^ to 10^−10^, a sufficient number of negative droplets were recorded and, hence calculation of the target concentration was feasible (Figure 3B). Figure 3C,D depict the linear regression line of the standard curve and the R^2^ value in qPCR (for 10 dilutions) and digital PCR (for five dilutions). The coefficient of correlation was 0.9998 for qPCR and 0.9999 for digital PCR. Figure 3E shows the copy numbers calculated in digital PCR assays and qPCR reactions (calculated according to the Prexcel-Q method) [27].

### 2.4. Analysis of Plant Samples Using Duplex qPCR and Singleplex ddPCR Assays

Lyophilized citrus samples were analyzed in duplicate assays by duplex qPCR (for CLas and COX genes) and by singleplex ddPCR (for CLas). Duplex ddPCR was not conducted since the droplets from COX reaction could not be counted unless the Ct values were 23 or higher. A total of 57 samples were positive by both qPCR and ddPCR. Results shown in Appendix A are from 32 samples that contained less than 50% positive droplets. However, since the copy number of the COX gene per cell was often high, ddPCR could not be conducted as a duplex reaction. In singleplex ddPCR reactions conducted without the reference gene, it is impossible to quantify the CLas copy numbers in relation to the number of host cells. 

### 2.5. Analysis of Samples from Diverse Taxa for CLas by qPCR Using COX and MDH as Reference Genes

Both COX and MDH were amplified from all 36 accessions belonging to 12 diverse taxa in duplex qPCR assays (Table 2). The Ct values for CLas positive samples were similar within each sample (approximately one Ct difference) with both assays in most cases (data not shown). The Ct values for COX ranged from 16 to 25 amongst different accessions, while the range for MDH for replicate samples ranged from 26 to 37. The Ct difference between COX and MDH for individual samples ranged from 8 to 16. Most citrus samples showed a Ct difference of 8–11. Maximum differences of 12–16 in Ct values between COX and MDH genes were observed in extractions made from actively growing plants such as *Bergera* and *Murraya*. When ten-fold dilutions of a CLas positive DNA were tested in duplex qPCR for CLas and COX or CLas and MDH in separate reactions, MDH was not detected in dilutions where the Ct value of COX was >29 (Table 3). 

### 2.6. Comparison of COX and MDH as Internal Controls for Detection of Two Target Genes of CLas by qPCR

Duplex qPCR reactions were carried out using ten-fold dilutions of a CLas positive citrus DNA template (prepared by diluting with DNA from a healthy citrus plant). The duplex reactions for CLas 16S rDNA were carried out along with primers and probes for COX or MDH. Since the dilutions contained approximately equal amounts of host DNA, all dilutions showed a Ct of around 18 for COX and 26 for MDH. A plot of Ct values for CLas with either COX or MDH at different dilutions is shown in Figure 4A. The Ct values for CLas in duplex reactions conducted with MDH primers and probe were about 1.5–2 Ct values lower than those conducted with COX primers and probe. At dilutions 10^−1^ to 10^−3^, the COX Ct range was about 28 to 36.

A similar set of reactions was carried out for detecting the RNR gene of CLas [29] using the serial dilutions of CLas infected citrus DNA (Figure 4B). The Ct values for RNR with MDH as the reference gene were about 0.9 to 1.8 Ct values lower than those obtained with COX as the reference gene. Average Ct values of triplicate reactions of the above experiments are shown in Appendix A. 

### 2.7. ddPCR of Reference Gene MDH and CLas in Singleplex and Duplex Assays

A CLas positive DNA preparation (Ct 24) was used to prepare ten-fold serial dilutions using TE buffer (10 mM Tris-HCl, 1 mM EDTA, pH 8.0). ddPCR assays were conducted to detect internal control gene MDH alone, MDH in the presence of CLas, CLas alone, and CLas in the presence of MDH. Figure 5A shows the number of positive droplets detected for MDH in a singleplex reaction and a duplex reaction (with MDH and CLas), CLas in a singleplex reaction, and a duplex reaction (with primers and probe for CLas and MDH) using different template dilutions. Linear regression lines are plotted in Figure 5B,C. The results show that the number of positive droplets in MDH samples and MDH + CLas are not significantly different. Similarly, MDH does not interfere with CLas detection when multiplex reactions are conducted. Coefficient of regression values (R^2^) was 0.9762 (MDH only); 0.9633 (MDH in duplex reaction); 0.9858 (CLas only), and 0.9977 (for CLas in duplex reaction). 

## 3. Discussion

### 3.1. HLB Situation in California 

The citrus industry in the USA has battled HLB for about 16 years, and management of the disease has proved to be challenging. After the first report of HLB in Brazil in 2004 [30,31] and Florida in 2005 [32], it was realized that the disease was already widespread in both regions, and eradication was not an option. Following the initial find of HLB in CA [4], various quarantine measures were enforced by the regulatory agency (California Department of Food & Agriculture), including restrictions on the movement of plant materials, vector control, intensive and repeated testing of all HLB hosts within a one-mile radius around the find site. These actions have been successful in disease mitigation, since only one additional CLas-positive tree (with the Hacienda Heights isolate of CLas) was found in the first find site in the next seven years. Hacienda Heights isolate has not been detected outside the area since the initial find in 2012, suggesting the possible eradication of this specific isolate. This is an excellent example of early detection of HLB and an effective disease management strategy. However, in subsequent years, additional foci of infection have been reported from non-commercial backyard citrus trees in Southern California. Based on sequencing studies, it was inferred that subsequent HLB finds in Southern California may be due to multiple populations of the pathogen [5] and probably caused by illegal introduction of the HLB infected plant material in the past.

In most cases, CLas-positive ACP led to the detection of compromised citrus trees that were HLB-positive but usually non-symptomatic. In Southern California, about 2400 CLas positive trees have been reported as of August 2021. Commercial citrus groves have been free of HLB, and it may be possible to delay the onset of the HLB epidemic if proper disease control strategies are practiced.

### 3.2. Importance of Detecting Samples with Low Titer of CLas

In California, monitoring for HLB is mainly done by qPCR assays of ACP and citrus samples. Neither citrus plants nor ACP carry the CLas bacterium throughout the year at high levels. ACP with presumably very low titers of CLas appear to cluster [33] and are generally regarded as inconclusive in HLB surveys. However, if these “hot spots” provide early warnings for new foci of infection, it is crucial that samples with high Ct from either ACP or plants are not overlooked in routine surveys. Fine-tuning of existing testing methods will help increase the sensitivity of the tests conducted to detect low titers of CLas.

### 3.3. Importance of Reference Genes

The amount of starting material, sample acquisition and handling, the integrity of the sample types utilized, tissues selected for analysis, sample processing methods, PCR inhibitors, reactions efficiency, and final data analysis can vary significantly among different laboratories and between experiments conducted at various times. Appropriate normalization of the data requires including validated reference genes in the detection systems [12,34]. Sample to sample variations are usually corrected by simultaneous amplification of an internal reference gene and the target gene [35]. Several programs such as NormFinder, BestKeeper, and geNorm are used to select reference genes for gene expression studies [36,37,38]. Rigorous guidelines for proper experimental conditions and meaningful evaluation of qPCR results are available for gene expression studies as described in MIQE guidelines (minimum information for publication of quantitative real-time PCR experiments) for qPCR [39], and for digital PCR [40]. Similar guidelines for selecting reference genes for DNA-based assays are needed to increase the inherent value of qPCR data. 

### 3.4. COX Is Suboptimal as a Reference Gene 

As a mitochondrial gene, COX can be present in high copy numbers in plant cells. In metabolically active, growing tissues such as shoot tips, the number of mitochondria per cell will be much higher than in mature tissues. Acquisition of CLas from the young flush tissue and subsequent transmission can lead to rapid disease spread through ACP feeding [8]. Detecting the pathogen in young and fast-growing tissues may be relevant for effective disease prevention and exclusion strategies. We have observed higher levels of COX gene in greenhouse-grown plants and in citrus relatives such as *Murraya paniculata* and *Bergera koenigii* that are highly preferred by ACP [41]. HLB surveys often include citrus relative genera, since disease spread can occur through ACP feeding on ornamental trees [42]. The inclusion of non-citrus host plants in HLB surveys appears necessary for an integrated disease management regimen that needs to be followed, especially at the beginning of an epidemic.

In most of our CLas assays using extractions from mature trees, Ct values for COX varied from about 20 to 23. However, in routine assays for CLas in extractions from greenhouse-grown plants and assays involving young tissue or bark, COX Ct values around 16–26 were documented. Excessive amounts of COX may overwhelm the PCR reaction and interfere with the detection of CLas. A comparison of COX and MDH as reference genes showed that when the Ct value of COX is 29 and above, there may not be enough nuclear DNA since the single-copy gene, MDH, is not detectable in parallel experiments. A cut-off value for COX at about Ct 27 may ensure that the extract used for CLas detection assays will have enough genomic DNA so that interpretation of the data will not be misleading. In this study, different taxa included in the analysis resulted in significant differences in Ct values between COX and MDH (about 8–16). DNA templates increase by one log every 3.322 cycles in optimal qPCR assays [43]. When high risk samples (from quarantine areas) are assayed, extractions with high Ct values for COX (>27) without adequate genomic DNA may lead to false-negative results. 

Ideally, the reference gene included in duplex assays should be present at an approximately similar level as the target gene [44]. Mitochondrial genes, by definition, do not satisfy this criterion since the number of mitochondria per cell is high and can vary significantly between tissue types. Higher levels of a reference gene may result in reagent depletion and may affect the amplification of the target gene leading to false negatives [43].

### 3.5. Selection of an Alternate Single Copy Universal Reference Gene

In the genome of *C. clementina*, there are three genes coding for three different MDH enzymes: EC 1.1.1.37, EC 1.1.1.39, and EC 1.1.1.40. The 906 bp sequence selected in this study is from the gene that codes for EC 1.1.1.40. This enzyme catalyzes the oxidative decarboxylation of malate in the presence of NADP+ into pyruvate, carbon dioxide and NADPH (https://www.genome.jp/dbget-bin/www_bget?ec:1.1.1.40 accessed on 2 October 2021). While this gene is 4257 bp long with 18 exons and 17 introns, the 900 bp fragment selected for this study contains four exons (Appendix A). The fragment of the MDH gene selected aligns to scaffold 1:7405845-7405936 (Phytozome 12.1, Citrus clementina v1.0), a 92 bp region represented in the genome only once indicating that it is a single copy gene. In the *Citrus sinensis* genome (Phytozome 12.1, Citrus sinensis v1.1 accessed on 2 October 2021), the selected region aligns to scaffold 00027: 654682-654773.

### 3.6. Reference Gene Requirement in Digital PCR

In our early ddPCR assays, we carried out singleplex assays for CLas, because in duplex assays COX oversaturated the droplets, and the need for an appropriate reference gene was obvious. While qPCR is an assay amenable to high throughput and has a broad dynamic detection range, ddPCR assays with a higher degree of sensitivity are valuable for absolute quantification and confirm positive samples that are considered ambiguous by qPCR. Accordingly, the reference gene is helpful to measure the amount of host DNA, while the target gene quantifies the pathogen. A mitochondrial gene such as COX will have a variable number of copies per cell from hundreds to a few thousand [19,20], and hence is not amenable to normalization and proper quantification. In HLB surveys, often qualitative detection of the associated pathogen is considered more critical than quantification. ddPCR may be useful as an alternative detection method for samples with low titers. qPCR reactions are affected by the presence of inhibitors, and certain citrus relatives can have a very high level of inhibitors. Depending on the extraction method and sample types (e.g., roots), inhibitors can significantly affect qPCR results. Compartmentalizing reaction mix into thousands of droplets (nanoliter quantities) reduces the negative effect of inhibitors. Hence, ddPCR may be an essential tool for diagnostic assays using samples that may have inhibitors hindering qPCR reactions. ddPCR can also be used for precise quantification of the target gene. Using MDH for normalizing the results, we have determined that in highly infected plant tissue with a Ct of 24, there will be about one CLas per 18–20 plant cells (Figure 5).

### 3.7. Importance of Appropriate Serial Dilutions

The linearity and dynamic range of sensitivity of duplex assays for CLas 16S rDNA, using either COX or MDH were analyzed using tenfold serial dilutions of DNA of a CLas positive sample. The dilutions were made using DNA from a CLas negative, healthy citrus extract. Dilutions in water or TE can be misleading since we are interested in detecting varying levels of CLas titer under natural conditions with about the same level of background host DNA. The use of MDH increased the sensitivity of detection of two target genes of CLas, 16S rDNA and RNR, by about 1–2 Ct values in most dilutions tested. Finally, we analyzed the linearity and sensitivity of both MDH and CLas (16S rDNA) in singleplex and duplex ddPCR assays. The results showed that the linearity and sensitivity of both assays were not affected by duplex reactions. 

## 4. Materials and Methods

### 4.1. Sequencing of a Single Copy Nuclear Gene from Members of Aurantioideae

About 900 bp region from MDH (nucleotide 7405482 to 7406388 in Scaffold 1 of the genome of *Citrus clementina*; NW_006263303.1) was analyzed from a wide variety of citrus cultivars and citrus relatives. A total of 197 plant accessions were sampled from the Citrus Variety Collection (http://www.citrusvariety.ucr.edu/citrus accessed on 2 October 2021), University of California, Riverside, CA. While most of these accessions belonged to the subfamily Aurantioideae to which *Citrus* belongs, a few accessions from other Rutaceae subfamilies were included in this study. Details of selected representative taxa are given in Appendix A. DNA from leaf tissue was extracted using Qiagen Plant DNeasy kit (Qiagen Inc., Germantown, MD, USA). PCR amplification of the 906 bp fragment of the nuclear MDH gene was conducted using primers CIT637 and CIT638 (Table 1), as described previously [22]. PCR was carried out in 50 µL reactions containing 20 mM Tris-HCl, pH 8.8 with 10 mM KCl, 10 mM (NH_4_)_2_SO_4_, 2 mM MgSO_4_, 0.1% Triton X-100, 0.2 µM of each primer, and 1.25 units of Taq DNA polymerase (New England Biolabs, Ipswich, MA, USA). Thermal cycling consisted of one cycle of initial denaturation at 95 °C, 3′, 35 cycles of denaturation at 95 °C, 15″, annealing at 55 °C, 15″, extension at 72 °C, 60″, followed by a final extension at 72 °C, for 5′. The PCR products were electrophoresed on agarose gels, purified using QiaExII kit (Qiagen Inc., Germantown, MD, USA), and cloned into pCR4 TOPO vector (Invitrogen, Waltham, MA, USA). The plasmids were purified and sequenced at the University of California, Riverside IIGB core facility (Sanger sequencing using Applied Biosystems H37306l DNA sequencer) using T7 and T3 universal primers. Haplotypes were deduced by analyzing sequences of 8–16 clones per accession. The sequences were aligned using ClustalX [23], and a cladogram was constructed using Maximum Parsimony method [24,25].

### 4.2. Design of Primers and Probes for Detecting Internal Control Gene, Malate Dehydrogenase

Conserved sequences of a 92 bp region of MDH gene were used to design forward and reverse primers, as well as a fluorescent probe suitable for qPCR and digital PCR assays (Figure 2). Primers and probes were synthesized by Integrated DNA technologies, Iowa, USA. Other primers and probes used in this study to amplify the 16S rDNA fragment of CLas and the citrus COX gene are shown in Table 1.

### 4.3. Standard Curves for qPCR and ddPCR Assays 

A 1167 bp region of 16S rDNA fragment of CLas was amplified by PCR from a CLas positive psyllid extraction with primers OI1 and OI2C primers [26] (modified as shown in Table 1) and cloned in a pGEM-T plasmid vector (Promega) as described earlier [6]. Ten-fold dilutions of plasmid DNA were used for generating standard curves in qPCR and droplet digital PCR (ddPCR) assays. The DNA concentration of the plasmid preparation was measured using a Nanodrop 2000c spectrophotometer and a Qubit 2.0 fluorometer (Thermo Fisher Scientific, Waltham, MA, USA). 

### 4.4. qPCR Assays and Analysis of Results

We used either Bio-Rad CFX96 real-time PCR system (Bio-Rad laboratories, Hercules, CA, USA) or an ABI ViiA7 real-time PCR system (Applied Biosystems, Waltham, MA, USA) for conducting qPCR assays. The general protocol used for qPCR was essentially as described earlier [7]. Briefly, the 20 ul qPCR reaction consisted of 2X Mastermix (SSO Advanced Universal Probes Supermix with reference dye Rox, from Bio-Rad) with appropriate primers and probe for singleplex or duplex reactions. The reaction mixes included gene-specific primers and probes for 16S rDNA with either COX or MDH gene (described above) for duplex reactions. The fluorescence signal generated is depicted as ΔRn (normalized reporter value Rn minus the base signal) plotted against cycle threshold value (Ct). Copy numbers obtained in qPCR standard curves were calculated using Prexcel-Q method [27].

### 4.5. Controls for qPCR and ddPCR Assays

DNA extractions from known CLas-positive trees from Fort Pierce, Florida were used to prepare ten-fold serial dilutions. DNA extractions from known CLas negative trees maintained in a greenhouse at the USDA facility in Riverside, CA were utilized for preparing serial dilutions for some experiments. Aliquots were prepared, stored at −20 °C and used as needed. All the DNA extractions were prepared using Plant MagAttract DNA extraction kit (Qiagen Inc., Germantown, MD, USA). These controls were used for qPCR and ddPCR assays, and as positive controls in all other diagnostic assays described in this study.

### 4.6. Comparison of COX and MDH as Internal Controls for Detection of CLas by qPCR

Ten-fold serial dilutions of a CLas positive extract were prepared in healthy citrus DNA extractions (as described before) and analyzed by duplex qPCR for 16S rDNA, using both COX [7] and MDH as internal controls in parallel assays. A second target gene, ribonuclease reductase (RNR) [29] of CLas was analyzed using COX and MDH as internal controls in separate reactions. 

### 4.7. Analysis of Samples from Diverse Taxa for Detection of CLas by qPCR Using COX and MDH as Reference Genes

Stored frozen DNA extractions from 36 accessions belonging to 12 diverse genera of Aurantioideae, retrieved from a previously conducted field trial in Fort Pierce, Florida were used for this study. These plants were planted in 2009, collected samples, and DNA extractions were tested for CLas multiple times over six years (Supplementary Table in [28]). Duplex qPCR assays were carried out for CLas 16S rDNA using COX and MDH as reference genes in parallel assays.

### 4.8. Analysis of Citrus Samples by Duplex qPCR and Singleplex ddPCR 

The samples used for this part of the study were a part of an evaluation panel prepared to compare different novel early detection technologies developed in CA for testing HLB-associated CLas from plants. Briefly, citrus samples were collected in Texas, lyophilized (using services of Quality Bioresources, Inc., Seguin, TX, USA), ground by OPS Diagnostics LLC, Lebanon, NJ, USA, and shipped to various participating laboratories (including our laboratory) as blind samples [45]. DNA extractions were prepared using a Plant MagAttract kit (Qiagen Inc., Germantown, MD, USA) in 96-well plates. Duplex qPCR reactions were carried out as described before. 

A 20 µL reaction was set up by mixing 10 µL of 2X ddPCR supermix (Bio-Rad), 900 nM of forward and reverse primers and 250 nM of FAM-labeled Taqman probe specific for 16S rDNA of CLas. ddPCRs were conducted using a Bio-Rad digital PCR system equipped with an automated droplet generator (QX200; Bio-Rad Laboratories, Hercules, CA, USA), as recommended by the manufacturer. Specialized DG32 cartridges and ddPCR oil designated for either Evagreen-based or, probe-based reactions (as appropriate) were loaded in the droplet generator. The reaction mix and oil were then mixed within the DG32 cartridges to create an oil-water emulsion containing up to 20,000 nanoliter-sized droplets. PCR plates containing the reaction mix were placed in the automated droplet generator, partitions were made through water-in-oil emulsion, and droplets were generated.

In our experiments, approximately 14,000–15,000 droplets were generated in each well. The PCR plates were sealed using a PX1 PCR plate sealer, and PCR was conducted in a Bio-Rad C1000 thermal cycler with the following thermal profile: initial denaturation at 95 °C for 10 min, 40 cycles of denaturation at 95 °C for 30 s, annealing/extension at 60 °C for one minute with a ramp rate of 2 °C/s between steps. After completion of PCR, endpoint data collected using a Bio-Rad QX200 droplet reader was analyzed using the Bio-Rad QuantaSoft™ program. Fluorescence measurements were recorded in the appropriate channel in the reader. The reader counted the number of droplets containing the target sequence (positive) and droplets without the target (negative). Each positive droplet was assigned a value of one and negative droplets were assigned a value of zero. A Poisson correction was applied by the QuantaSoft™ program used for analysis so that the mean number of target sequences per partition can be estimated. Concentration reported after the analysis represents the number of target copies/uL of the template.

### 4.9. Duplex ddPCR Reactions

We used COX or MDH as reference genes and 16s and RNR genes of CLas as test genes in duplex PCR reactions. The TaqMan probes consisted of fluorophores, FAM (for CLas probe), and HEX (for COX and MDH probes). Various concentrations of primers and probes were evaluated in standardization assays (data not shown). Fluorescence measurements of droplets were recorded in both channels in the reader for duplex assays. Ten-fold serial dilutions of selected samples were prepared and analyzed by qPCR, using MDH as an internal control in singleplex and duplex ddPCR reactions to analyze linearity in duplex reactions.

## 5. Conclusions

We have developed an improved reference gene, MDH, suitable in duplex assays designed to detect CLas from plants. The detection system for the reference gene developed in this study is expected to be universal for all known HLB hosts in the subfamily, Aurantioideae. The Mitochondrial COX gene that is commonly used as a reference gene in duplex qPCRs is not adequate, since the copy number varies depending on the metabolic state of the tissue, growth conditions, and plant type. Using laboratory assays, we have demonstrated the linearity and sensitivity of detection in duplex qPCR and ddPCR assays when MDH is used as the reference gene. Data normalization requires a stable reference gene that does not overwhelm the PCR reaction due to its copy number. There is a need to develop standards for reference genes in both qPCR and ddPCR assays similar to MIQE guidelines, used mainly in gene expression studies. MDH will be useful as a reference gene in diagnostic qPCR assays used to detect many citrus pathogens.

## Figures and Tables

**Figure 1 plants-10-02111-f001:**
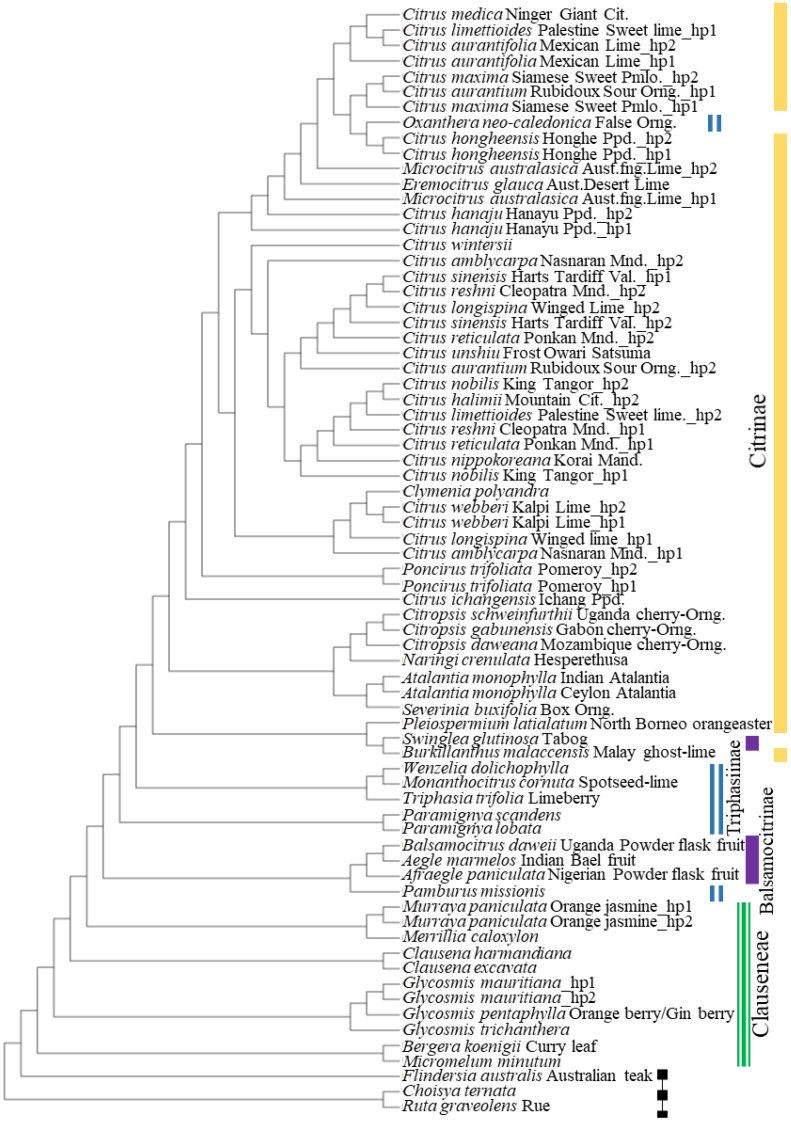
Cladogram constructed using 900 bp sequence of malate dehydrogenase gene fragment. The Maximum Parsimony (MP) tree constructed using MEGA X had a length of 851, consistency index was 0.7529 (0.6366), the retention index was 0.8069 (0.8069), and the composite index was 0.6076 (0.5137) for all sites and for parsimony-informative sites (in parentheses). The MP tree was obtained using the Subtree-Pruning-Regrafting algorithm. The aligned file with 72 sequences had a total of 954 positions in the final dataset. Clusters representing major groups of Aurantioideae are indicated. Three non-Aurantioid taxa, *Flindersia australis*, *Choisya ternata* and *Ruta graveolens* formed the outgroup for the cladogram. Common names for taxa are shown next to the scientific names. Hp1 and hp2 indicate haplotypes.

**Figure 2 plants-10-02111-f002:**
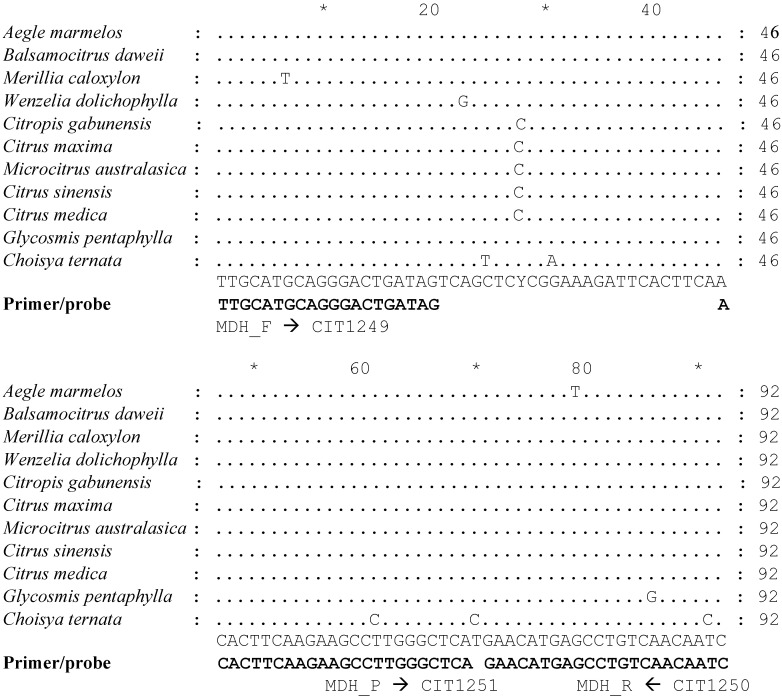
Primers and probe for amplification and detection of a 92 bp gene fragment of MDH reference gene from taxa of the subfamily Aurantioideae. The “*” sign indicates 10 base count.

**Figure 3 plants-10-02111-f003:**
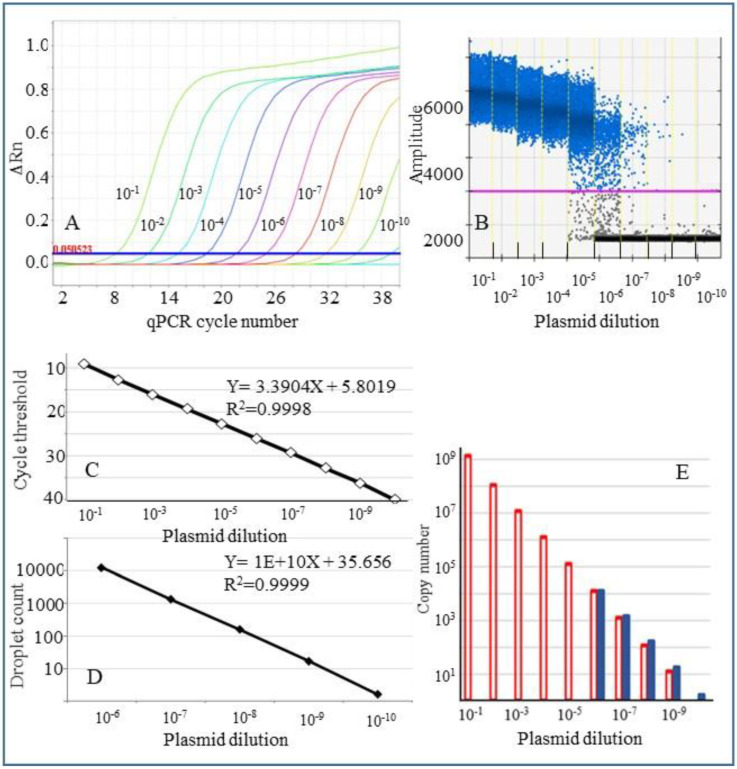
Linear regression analysis of qPCR and ddPCR conducted using ten-fold serial dilutions of a plasmid preparation containing CLas 16S rDNA fragment. (**A**). Amplification plots of plasmid serial dilutions in qPCR. ΔRn is the normalized reporter value (Rn) minus the Rn value of the baseline signal. Cycle threshold values are indicated on X-axis. The plasmid dilutions are shown to the left of each amplification graph. (**B**). ddPCR results of the plasmid dilutions used in qPCR. About 14,000–16,000 droplets passed QC in each assay. A manually determined cut-off line separated the distribution of negative and positive droplets shown. In the five lower dilutions (10^−1^ to 10^−5^), the droplets appear completely saturated (no negative droplets), and hence these are not ideal for quantification. (**C**,**D**). Linear regression line calculated based on the data points is indicated for qPCR (**C**) and ddPCR (**D**). (**E**). Copy numbers for qPCR (open, red bars) and digital PCR (solid, blue bars) calculated as described in materials and methods.

**Figure 4 plants-10-02111-f004:**
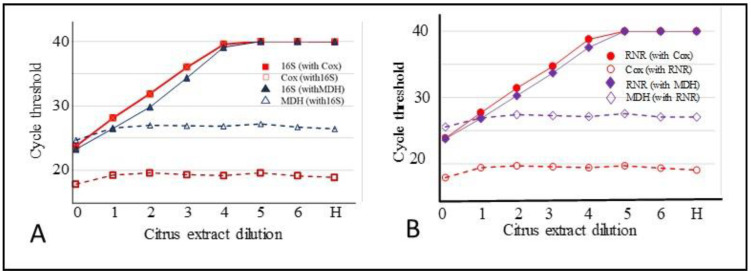
(**A**) Comparison of qPCR data in duplex assays amplifying 16S rDNA of CLas along with two separate reference genes, COX or MDH. (**B**). Comparison of qPCR amplification of CLas RNR gene along with COX or MDH in duplex reactions. Healthy citrus extracts were used to dilute CLas positive extracts. The CLas positive citrus extraction was diluted using CLas negative citrus extraction (H) as a diluent resulting in a near-constant Ct value of COX (about 18) and for MDH (about 26) in all dilutions. Ct values for the reference genes were approximately constant across dilutions. Ct value of 40 indicates a lack of amplification.

**Figure 5 plants-10-02111-f005:**
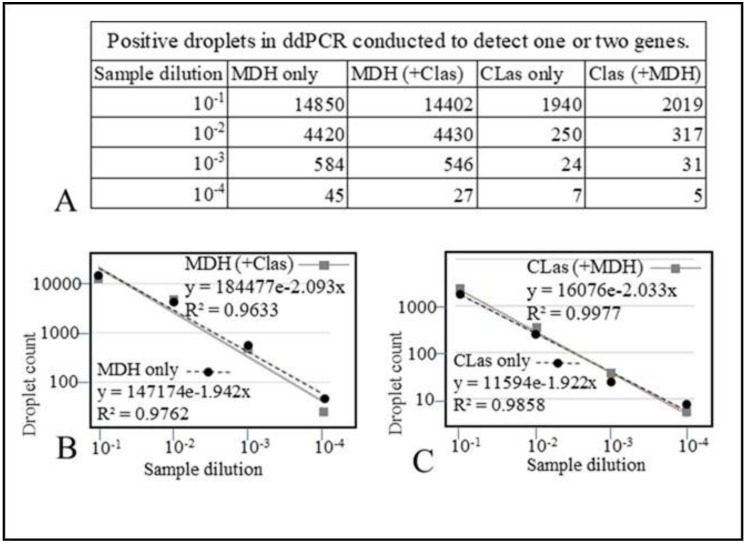
ddPCR of CLas and MDH in singleplex and duplex assays. Serial dilutions of a CLas positive citrus DNA extract were made using TE as a diluent. Columns 2 and 3 in panel (**A**) show number of positive droplets for MDH assay when conducted alone or with CLas. Columns 4 and 5 show the number of droplets recorded as positive for CLas in singleplex or duplex assays (conducted with MDH). Panel (**B**) shows the regression line for MDH (conducted with or without CLas). Panel (**C**) shows regression analysis for CLas conducted with or without MDH.

**Table 1 plants-10-02111-t001:** Primers used for conventional, qPCR, and digital PCR detection of various targets. Primers cit637 and cit638 were used for conventional PCR to amplify 900 bp fragments of MDH gene. Other primers and probes were used for qPCR and ddPCR reactions.

Primer	Sequence (5′ to 3′)	Target	Reference
CIT205a	GCGCGTATGCgAATACGAGCGGCA	CLas 16s forward (OA1) *	[26]
CIT206	GCCTCGCGACTTCGCAACCCAT	CLas 16s reverse (OIc)	
CIT295a	TCGAGCGCGTATGCgAATACG	Clas-16s forward *	[7]
CIT298	TGCGTTATCCCGTAGAAAAAGGTAG	Clas-16s reverse	
CIT301	CAGACGGGTGAGTAACGCG	Clas-16s probe	
CIT315	GTATGCCACGTCGCATTCCAGA	COX-F forward	[7]
CIT316	ATCCAGATGCTTACGCTGG	COX-P probe **	
CIT317	GCCAAAACTGCTAAGGGCATTC	COX-R reverse	
CIT637	GCTCCTGTGGAAGAGACCC	MDH forward	This study
CIT638	GCTCCAGAGATGACCAAAC	MDH reverse	
CIT1249	TTGCATGCAGGGACTGATAG	MDH forward	This study
CIT1250	GATTGTTGACAGGCTCATGTTC	MDH reverse	
CIT1251	ACACTTCAAGAAGCCTTGGGCTCA	MDH Probe **	

* Original primer sequence from [7,26] were modified, and extra bases (indicated by lower case letter) were added to primers cit205a cit295a. ** Fluorophore JOE™ was used for assays conducted in the ABI ViiA7 real-time PCR machine, and HEX™ was used for BioRad CFX96 qPCR and BioRad digital droplet PCR instruments.

**Table 2 plants-10-02111-t002:** Evaluation of two reference genes, COX and MDH, in duplex qPCR assays. Two duplex qPCR assays for detecting 16S rDNA of CLas were conducted, one using variable multicopy mitochondrial COX gene and another using nuclear single copy MDH gene. The plant samples were collected from a field in HLB endemic area in Florida [28], DNA extractions were made and used for assays. The cycle threshold (Ct) values for only the reference genes are shown. In specific citrus relative genera, a Ct difference of up to 15 was observed between the two reference genes. The CRC numbers refer to the citrus research center IDs for the seed source trees from the Citrus Variety Collection, University of California Riverside.

CRC#	Genus	Species	Variety	COX	MDH	Difference
3564	*Citrus*	*lycopersicaformis*	Kokni Orange	19.05	27.17	8.12
3546	*Citrus*	*medica*	South Coast Field Station	18.26	26.57	8.31
3673	*Microcitrus*	*australis*	Australian round lime	17.77	26.19	8.42
1482	*Citrus*	*limettioides*	Palestine sweet lime	17.95	26.65	8.70
3474	*Citrus*	*intermedia*	Yama-mikan sour orange	17.98	26.79	8.81
3149	*Citrus*	*benikoji*	Unnamed tangor	19.53	28.42	8.89
3514	*Balsamocitrus*	*daweii*	Uganda powder flask	18.52	27.58	9.06
3052	*Citrus*	*latipes*	Khasi papeda	18.56	27.93	9.37
4107	*Severinia*	*buxifolia*	Chinese box orange	18.49	27.94	9.45
1484	*Microcitrus*	*australasica*	Australian finger lime	18.70	28.25	9.54
3225	*Citrus*	*maderaspatana*	Kitchli Sour orange hybrid	18.62	28.24	9.62
1455	*Citrus*	*webberi*	Kalpi papeda	18.28	28.05	9.77
3752	*Citrus*	*reticulata*	Som Keowan mandarin	16.56	26.40	9.84
1485	*Microcitrus*	*hybrid*	Sydney Hybrid	20.47	30.32	9.85
3771	X *Citroncirus*	sp.	Swingle citrumelo	17.79	27.78	9.99
4105	*Eremocitrus*	*glauca*	Australian Desert lime	18.82	28.83	10.01
2588	*Citrus*	*taiwanica*	Nansho daidai sour orange	18.06	28.08	10.02
3959	*Citrus*	*maxima*	Egami Buntan pummelo	18.68	28.75	10.08
3842	*Citrus*	*celebica*	Alemow hybrid	17.95	28.14	10.19
3957	X *Citroncirus*	sp.	X639 trifoliate hybrid	16.83	27.04	10.20
2320	*Citrus*	*longispina*	Winged lime	18.64	28.90	10.26
2317	*Citrus*	*excelsa*	Limon Real papeda	19.37	29.64	10.26
2485	*Citrus*	*amblycarpa*	Nasnaran mandarin	17.56	27.89	10.33
3147	*Citrus*	*leiocarpa*	Koji mandarin	19.55	29.98	10.42
3907	*Citrus*	*hassaku*	Hassaku pummelo hybrid	17.84	28.27	10.43
1491	*Severinia*	*buxifolia*	Chinese box orange	19.13	29.80	10.67
2427	*Citrus*	*davaoensis*	Davao lemon	16.28	26.99	10.71
3549	*Poncirus*	*trifoliata*	Simmons trifoliate	18.72	29.60	10.89
4007	*Poncirus*	*trifoliata*	Little-Leaf trifoliate	16.68	27.62	10.93
2879	*Hesperethusa*	*crenulata*	Hesperethusa	18.87	29.83	10.96
301	X *Citroncirus*	sp.	Rusk citrange	17.60	28.91	11.31
3285	*Glycosmis*	*pentaphylla*	Orange berry	24.98	36.51	11.53
2878	*Aeglopsis*	*chevalieri*	Chevalier’s Aeglopsis	19.98	31.92	11.95
3165	*Bergera*	*koenigii*	Curry leaf	18.89	31.23	12.34
1637	*Murraya*	*paniculata*	Orange Jessamine	18.30	33.74	15.44
3171	*Murraya*	*paniculata*	Hawaiian Mock Orange	16.94	32.70	15.76

**Table 3 plants-10-02111-t003:** Citrus DNA extracts assayed for COX and MDH reveal approximately 6–8 Ct values lower for COX. Ten-fold serial dilutions of a citrus DNA extract (#631) were utilized for qPCRs, and Ct values are shown. No template control (NTC) was used as a negative control.

Dilution	COX Ct	MDH Ct
10^−1^	22.27	28.81
10^−2^	25.74	32.97
10^−3^	29.36	36.97
10^−4^	33.49	0
10^−5^	36.38	0
10^−6^	0	0
NTC	0	0

## Data Availability

The data presented in this study are available in the article or linked Appendix A.

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
