# Peer review of "An Improved Reference Gene for Detection of “Candidatus Liberibacter asiaticus” Associated with Citrus Huanglongbing by qPCR and Digital Droplet PCR Assays"

_plants, 2021, doi:10.3390/plants10102111_

Round 1
Reviewer 1 Report
The manuscript provides interesting data on the use of MDH as reference gene, in duplex assays to detect CLas from plants instead of the Mitochondrial COX gene that is commonly used as a reference gene in duplex qPCRs.
Keep in mind that HLB has caused significant financial damage to citrus industries worldwide it is essential to have a tool for an early detection of the disease and that it also avoids false negatives.
The manuscript is written in clear English, even if I am not mother tongue then I cannot judge this aspect. The introduction provides comprehensive information on the state of the art of the subject matter, as well as demonstrates the importance of having an alternate reference gene that has a low copy number in citrus hosts.
The cited literature is relevant and well referenced. The techniques and parameters evaluated are very satisfactory, as well as the inclusion in the study of species related to the Citrus genus, which are often important sources of inoculum. Moreover, the authors explain very well the results and correlating them.
This study offers an interesting starting to research and deepen the studies about the appropriate normalization of the data requires including validated reference genes in the detection systems also in other species.
Finally, I congratulate the authors for their work and my recommendation is to accept the manuscript for publication in present form.
Author Response
Reviewer #1 did not suggest any edits.
Reviewer 2 Report
This is a modest paper on an important disease
Author Response
Reviewer #2 did not suggest any edits.
Reviewer 3 Report
Citrus huanglongbing (HLB) disease associated with the alphaproteobacterium 'Candidatus Liberibacter asiaticus’ (CLas) causes significant financial damage to citrus industries worldwide. Therefore the early detection of CLas in citrus trees and citrus-related species infected by CLas is essential. In this study the authors have evaluated a novel nuclear single copy gene, MDH, in order to be used as a reference gene in duplex qPCR and ddPCR in place of the mitochondrial COX gene for the accurate and sensitive detection of CLas in citrus and citrus-related species.
The manuscript is well written, the results are well described and overall merits publication in Plants. Some minor suggestions/comments are found in the attached file.

Author Response
- All minor edits suggested by Reviewer #3 were incorporated [Ln 201 (198); Ln 253/254 (251); Ln 287 (284); Ln 360 (357); Ln 429 (426)].
- Ln 310-313 (307-310) were removed and moved to the beginning of section 2.1 as suggested. The next underlined sentence was slightly modified for proper context.
- Please note that the line numbers provided here are from the edited/re-submitted version along with the line numbers from the original submission in parenthesis.